# OpenReview forum: "MATH-Beyond: A Benchmark for RL to Expand Beyond the Base Model"
_ICLR.cc/2026/Conference — ICLR 2026 Poster_

### Official Review · Reviewer_aWPb · 2025-10-22

**Soundness:** 2
**Presentation:** 3
**Contribution:** 1
**Rating:** 2
**Confidence:** 5

**Summary:**

This paper introduces MATH-B, a benchmark designed to evaluate whether reasoning models fine-tuned with RL can genuinely “expand” beyond the capabilities of their base models, aiming to push progress past the current plateau. The benchmark is constructed by filtering existing datasets, DAPO-Math-17K and DeepScaleR, to retain problems unsolved by several open-source models even under a large sampling budget (pass@1024). The authors define an “Expansion Rate” metric, effectively pass@k on a zero-baseline set, and evaluate several RL-trained and SFT models. They find that most RL methods yield only modest (<10%) improvements, whereas long-chain-of-thought (CoT) distillation achieves substantially higher (>50%) gains. The authors argue that MATH-B can serve as a diagnostic tool to guide future exploration-driven RL research.

**Strengths:**

1. The limits of RL-fine-tuned reasoning models is an important topic, especially given saturation on common benchmarks.

2. The paper is reasonably well written and clearly explains its data-filtering pipeline and evaluation setup. The figures and tables (e.g., dataset construction flow, pass@k comparisons) are well organized.

**Weaknesses:**

1. The dataset is almost entirely derived from existing public datasets (DAPO-Math-17K, DeepScaleR), using common filtering, for example, deduplication, sampling, verification, etc. The resulting benchmark is a subset of known problems already curated and verified in prior works. The paper does not introduce new problem formulations, annotation schemes, or evaluation methodologies that go beyond what prior datasets or analyses already offer, especially given the availability of broader, harder, and more rigorously verified benchmarks such as MathArena [3], UQ [4] and more.

2. The motivation and methodology closely mirror those of DAPO [1] and DeepScaleR [2], both of which already include difficult math subsets, verification studies, and RL vs. SFT comparisons. These prior works went further by introducing RL frameworks, conducting ablations, and analyzing exploration effects

3. The evaluation simply confirms what is already known, that current RL methods (e.g., GRPO, R1-style) yield limited improvement over base models, especially on problems where base model can barely solve.

4. The authors claim MATH-B encourages “exploration-driven RL,” and can "move beyond the current plateau", but provide no evidences show that training with the dataset help model learn new reasoning strategies. Thus, its value as a benchmark remains speculative.

[1] Yu, Qiying, et al. "Dapo: An open-source llm reinforcement learning system at scale." arXiv preprint arXiv:2503.14476 (2025).

[2] Luo, Michael, et al. DeepScaleR: Surpassing O1-Preview with a 1.5B Model by Scaling RL. 2025, pretty-radio-b75.notion.site/DeepScaleR-Surpassing-O1-Preview-with-a-1-5B-Model-by-Scaling-RL-19681902c1468005bed8ca303013a4e2.

[3] Balunović, Mislav, et al. "Matharena: Evaluating llms on uncontaminated math competitions." arXiv preprint arXiv:2505.23281 (2025).

[4] Nie F, Liu K Z, Wang Z, et al. UQ: Assessing Language Models on Unsolved Questions[J]. arXiv preprint arXiv:2508.17580, 2025.

**Questions:**

1. In Figure 3, the authors show that the problems models find challenging are not necessarily those that humans typically struggle with. Why does this mismatch occur, and what specific failure modes of LLMs lead to zero-success rates across multiple models?

2. For problems outside MATH-B that models solve with very low probability, is success mainly due to genuine reasoning and correct problem-solving, or does it mostly arise from randomness, where the reasoning is incorrect but the final answer happens to match?

3. Given the reliance on GPT-5 for verification, how robust are the correctness checks? Did the model ever mis-verify solutions?

---

> ### Author Response · Authors · 2025-11-19
>
> Thank you for taking the time to review our work. We address each of the concerns below.
>
> **[W1, W2 – On similarities with datasets, pipelines, and contributions]**: The purpose of our work is to build model-specific test sets where the base models have pass@1024 ≈ 0. This enables researchers to readily measure whether an RL or exploration method (trained elsewhere) can genuinely expand a models’ reasoning boundary. Neither DAPO nor DeepScaleR nor MathArena nor UQ offer this functionality. Their aim is to evaluate general reasoning ability, not capability expansion. All the models that we consider obtain non-zero pass@1 and very high pass@1024 on AIME24/25, MATH-500, MathArena, etc [1–3]. This makes them unsuitable for diagnosing whether a post-training method actually expands what a model can solve.
>
> One could attempt to create capability-expansion subsets by filtering AIME24/25 using pass@1024 evaluations, but because many base models already have extremely high pass@1024 on these datasets, the resulting unsolved subsets would be too small to draw meaningful conclusions [1]. In contrast, MATH-B has over 100 problems per model with pass@1024 ≈ 0.
>
> Our pipeline intentionally reuses established cleaning and verification procedures. We view this as a strength of our work. We also novelly include a pass@1024 filtering stage, which enables MATH-B to fulfill its purpose of measuring capability expansion,  unlike existing datasets.
> Finally, our work targets academic-scale RL research. Frontier-scale datasets like UQ are intentionally much harder and outside the scope of what the models we study (≤8B) can realistically solve. Since most research on post-training considers ≤8B models (except for a handful of industry labs), MATH-B is practically relevant to a large share of the RL/exploration research community.
>
> **[W3 – On RL not yielding improvements and “no evidence for exploration”]:** Table 2 clearly shows that stronger RL recipes (Skywork) and increases in RL compute (Nemotron) both improve performance on MATH-B. These are precise examples of exploration or RL methods expanding a model’s boundary and therefore direct evidence that MATH-B can detect gains from RL training. The reviewer’s statement that we provide “no evidence that training with the dataset helps the model learn new reasoning strategies” rests on a misunderstanding: MATH-B is a test set, not a training set. While some high-level trends about RL vs SFT have been observed before, our contribution is to provide a benchmark that cleanly diagnoses capability expansion and can support methodological progress.
>
>
> **[Q1 – On human vs machine difficulty and failure modes of MATH-B]:** A mismatch between human difficulty and model difficulty is expected and well-documented. LLMs are known to fail on distributional variations that humans find trivial: high-digit arithmetic, symbolic perturbations, unusual factorization paths, or small structural rewrites [4–6]. These failures are governed by the model’s training distribution rather than human perception of problem hardness. In constructing MATH-B, we filtered for problems that several base models fail at pass@1024, regardless of human difficulty. We inspected the hardest problems qualitatively and did not observe a single unifying semantic pattern, which is consistent with prior work showing that model difficulty often does not map cleanly onto human notions of difficulty [4-6]. This is an inherent property of LLMs, not an issue with the benchmark.
>
> **[Q2 – On the correctness of CoT]:** The concern that rare successes may be due to random answer-matching rather than genuine reasoning has already been examined in prior work [1]. There they perform a detailed manual inspection of chain-of-thought traces for correct answers on the hardest problems (≤5% accuracy) for Qwen2.5-7B-Base and an RL-trained variant. They find that most correct outputs contain at least one valid reasoning trajectory, showing that low-probability success is typically grounded in real reasoning steps rather than chance. Since the models we evaluate belong to the same families (Qwen2.5, Qwen3, R1-Distill-Qwen2.5, etc.), there is no reason to expect a qualitative change in this behavior. Occasional lucky hits are always possible, but the available evidence indicates that correct solutions predominantly arise from genuine reasoning rather than randomness.

---

> ### Author Response · Authors · 2025-11-19
>
> **[Q3 – On verification of MATH-B]:** MATH-B is constructed by subsampling from DeepScaleR [7] and DAPO-Math-17K [8]. DeepScaleR contains AIME problems (1984–2023), AMC problems, and problems drawn from Omni-MATH, all of which are human-written and human-verified [7]. DAPO-Math-17K was collected from official contest sources and manually processed. All of these datasets have already gone through different forms of human verification. In addition to this, we run all MATH-B problems through two strong frontier-level models (o4-mini-high and GPT-5-Mini) for correctness verification. Further, multiple post-training methods improve performance on the test set (Table 2), further indicating the correctness of the dataset. While it is possible that o4-mini-high and GPT-5-Mini occasionally mis-verifies, the combination of human-origin problems, two independent frontier-model verifications, and successful post-training improvements makes systematic verification errors unlikely.
>
> We hope these clarifications address your concerns fully and help raise your confidence and rating.
>
> References:
>
> [1] Yue et al., “Does Reinforcement Learning Really Incentivize Reasoning Capacity in LLMs Beyond the Base Model?,” 2025
>
> [2] Hochlehnert et al., “A Sober Look at Progress in Language Model Reasoning: Pitfalls and Paths to Reproducibility?,” 2025
>
> [3] Yang et al., “Qwen3 Technical Report,” 2025
>
> [4] Huang et al., “MATH-Perturb: Benchmarking LLMs' Math Reasoning Abilities against Hard Perturbations,” 2025
>
> [5] Dziri et al., “Faith and Fate: Limits of Transformers on Compositionality,” 2023
>
> [6] Mirzadeh et al., “GSM-Symbolic: Understanding the Limitations of Mathematical Reasoning in Large Language Models,” 2025
>
> [7] Luo et al., “DeepScaleR: Surpassing O1-Preview with a 1.5B Model by Scaling RL,” 2025
>
> [8] Yu et al., “DAPO: An Open-Source LLM Reinforcement Learning System at Scale,” 2025

---

### Official Review · Reviewer_VmSx · 2025-10-31

**Soundness:** 3
**Presentation:** 3
**Contribution:** 3
**Rating:** 4
**Confidence:** 5

**Summary:**

1.Improving performance is explored on the proposed benchmark via RL requires methods that learn to reason in ways that go beyond base model capabilities in repeated sampling. Since the problems are drawn from subsets of DAPO-Math-17K and DeepScaleR datasets, they remain topically equivalent to standard high-school math.

**Strengths:**

1.The dataset is hard for 1.5b llm to follow, RL fine-tuned models such as Nemotron-Research-Reasoning-Qwen-1.5B and DeepScaleR-1.5B-Preview perform poorly on MATH-B at pass@1024.

**Weaknesses:**

1.The proposed dataset does not show the goodness on the performance rising after using the dataset for RL training.
2.The difficulity and other flag is not given for the dataset such as AIME24 and AIME25.
3.The quality of the proposed dataset from the LLM-based algorithm is not double-check on human experts.
4.These is no proof to show a variety of LLMs improve their reasoning ability on the application of the proposed dataset.
5.There is no comparison with the proposed dataset and other famous reasoning math dataset such as MATH-0500, AIME24, AIME25.

**Questions:**

1.Why does the proposed dataset fail to demonstrate performance gains when used for reinforcement learning (RL) training?
2.Why are difficulty levels and other metadata flags not provided for datasets such as AIME24 and AIME25?
3.Why is the quality of the proposed dataset, generated via an LLM-based algorithm, not validated through human expert review?
4.Why is there no evidence showing that a diverse set of LLMs improve their reasoning abilities when trained on the proposed dataset?
5.Why is there no comparative analysis between the proposed dataset and established reasoning math datasets such as MATH-500, AIME24, and AIME25?

---

> ### Author Response · Authors · 2025-11-19
>
> Thank you for taking the time to review our work. We address each of the concerns below.
>
> **[Main Contribution]:** We would first like to clarify the central aim of our paper, as there seems to be a misunderstanding. Prior work has shown that most RL methods mainly sharpen a base model’s existing reasoning paths rather than discover new ones [1]. However, this limitation is easy to overlook because widely used math benchmarks are already solved by base models at high pass@k [1]. Our goal is therefore to construct model-specific test sets where the base models have pass@1024 ≈ 0, so that one can reliably evaluate whether a post-training RL or exploration method (trained on some other dataset) genuinely expands the base model’s capabilities. The contribution is a test benchmark for measuring expansion, not a training dataset.
>
> **[W1, W4 & Q1, Q4 – On performance not improving after RL training]:** With this framing, it appears the reviewer has misunderstood the purpose of the dataset. MATH-B is a test set, not a training set. It is not intended to be used for RL training, so the concern that “dataset fails to demonstrate performance gains when used for reinforcement learning (RL) training” does not apply. For any base model, the correct usage is: (1) select the subset of MATH-B problems the base model fails at (pass@1024 ≈ 0), and (2) evaluate whether an RL or exploration method trained on another dataset expands that boundary. As shown in Table 2, several post-training methods do actually improve performance on MATH-B. We kindly request the reviewer to re-examine our work with this intended usage in mind.
>
> **[W5 & Q5 – On comparisons with MATH-500, AIME24, AIME25]:** MATH-500, AIME24, and AIME25 are designed to measure general math ability of models. They are not constructed to measure capability expansion of RL/exploration methods. All considered models achieve non-zero pass@1 on MATH-500/AIME24/AIME25 (see [1–3]), and their pass@1024 values are extremely high [1]. By contrast, every model we consider has a pass@1024 ≈ 0 on its corresponding MATH-B subset. The purpose of MATH-B is therefore different from MATH-500/AIME24/AIME25: our test set probes the extent to which  RL/exploration methods push a model beyond its existing boundary. We have nonetheless updated Figure 1 to include pass@1024 values on AIME24 from [1], and Table 2 now shows pass@1 values on AIME24 for all considered models. These additions further highlight the difficulty gap and why direct comparisons are not especially informative.
>
>
> **[W3 & Q3 – On human verification]:** MATH-B is constructed by subsampling from DeepScaleR [4] and DAPO-Math-17K [5]. DeepScaleR contains AIME problems (1984–2023), AMC problems, and problems drawn from Omni-MATH, all of which are human-written and human-verified. DAPO-Math-17K was collected from official contest sources and manually processed. Thus, our source data has already gone through different forms of human verification. In addition, we run all MATH-B problems through two strong frontier-level models (o4-mini-high and GPT-5-Mini) for correctness verification. Further, multiple post-training methods improve performance on the test set (Table 2), indicating the correctness of the dataset. We therefore believe additional human verification would be redundant.
>
> **[W2 & Q2 – Difficulty labels for AIME24/25]:** Our contribution is the construction and analysis of MATH-B, not the re-annotation of existing benchmarks. We include human difficulty labels for MATH-B only to illustrate that human difficulty often diverges from machine difficulty. LLMs have well-documented distributional failure modes: high-digit arithmetic, symbolic perturbations, or small variants of known problems [6–8]. Our results continue this line of evidence by showing that models can fail on problems that are topically ubiquitous high school problems. Providing difficulty labels for AIME24/25 does not meaningfully advance our contribution, so we opted not to include them.
>
> We hope these clarifications address your concerns fully and help raise your confidence and rating.
>
> References:
>
> [1] Yue et al., “Does Reinforcement Learning Really Incentivize Reasoning Capacity in LLMs Beyond the Base Model?,” 2025
>
> [2] Hochlehnert et al., “A Sober Look at Progress in Language Model Reasoning: Pitfalls and Paths to Reproducibility?,” 2025
>
> [3] Yang et al., “Qwen3 Technical Report,” 2025
>
> [4] Luo et al., “DeepScaleR: Surpassing O1-Preview with a 1.5B Model by Scaling RL,” 2025
>
> [5] Yu et al., “DAPO: An Open-Source LLM Reinforcement Learning System at Scale,” 2025
>
> [6] Huang et al., “MATH-Perturb: Benchmarking LLMs' Math Reasoning Abilities against Hard Perturbations,” 2025
>
> [7] Dziri et al., “Faith and Fate: Limits of Transformers on Compositionality,” 2023
>
> [8] Mirzadeh et al., “GSM-Symbolic: Understanding the Limitations of Mathematical Reasoning in Large Language Models,” 2025

---

### Official Review · Reviewer_uHA5 · 2025-11-01

**Soundness:** 2
**Presentation:** 3
**Contribution:** 2
**Rating:** 4
**Confidence:** 3

**Summary:**

This paper introduces MATH-Beyond (MATH-B), a diagnostic benchmark designed to evaluate whether post-training methods genuinely expand mathematical reasoning capabilities or merely refine existing ones. Observing that base models with large sampling budgets (pass@1024) already solve nearly all problems on widely used benchmarks, the authors construct MATH-B to challenge open-source models up to 8B parameters even under large sampling budgets. MATH-B serves as a "zero-baseline" diagnostic tool to catalyze research into exploration-driven RL approaches that achieve genuine capability expansion.

**Strengths:**

1. This paper addresses an important problem in post-training research, evaluating whether models genuinely explore new reasoning paths versus merely optimizing existing solution modes. This distinction between capability expansion and capability refinement is crucial for understanding the true progress in mathematical reasoning.
2. This presents a structured approach to dataset construction using automated methods, which could potentially improve efficiency compared to pure manual annotation.
3. The paper provides extensive experimental validation across multiple RL fine-tuned models and large sampling budgets to demonstrate the benchmark's effectiveness. The empirical results convincingly show that existing post-training methods struggle on MATH-B, validating the benchmark's design as a diagnostic tool.
4. The paper is well-written with clear motivation and logical flow.

**Weaknesses:**

1. This paper uses models with <8B parameters for data filtering may introduce model-specific biases. The filtering model's capability ceiling and training data preferences could propagate into the dataset, potentially affecting the benchmark's reliability and fairness across different model architectures and training paradigms.
2. The paper only evaluates models ≤8B parameters, leaving the benchmark's effectiveness for larger models (>8B, especially >70B) unverified. Without validation on state-of-the-art large models, it remains unclear whether MATH-B maintains discriminative power or exhibits ceiling effects for models that mainstream research focuses on, potentially limiting the benchmark's practical utility.
3. Evaluating the same or similar models that were used for data filtering may introduce circular dependencies and bias.

**Questions:**

1. How do you ensure that MATH-B problems require genuine reasoning exploration rather than reflecting the specific failure modes of the filtering models?
2. What are the results on large models (>8B), and does the benchmark maintain discriminative power at scale?

---

> ### Author Response · Authors · 2025-11-17
>
> Thank you for taking the time to read our paper and for the feedback.  We address each of your concerns below:
>
> **[W1 & W3 – On model-specific and circular biases]:** We would first like to clarify the design philosophy of MATH-B. The benchmark is intentionally model specific: it is built from the empirical failures of a given set of models, and its purpose is to evaluate post-training methods (e.g., a new RL algorithm), rather than post-trained models as they are. Our proposed benchmark measures whether a post-training method genuinely expands a base model’s capability to solve problems that were previously unsolved at pass@1024 ≈ 0. This setup is standard in machine learning. Benchmarks like ImageNet-A [1] and the whole line of work on adversarial examples [2] follow the same idea of identifying model weaknesses to drive methodological progress. Further, model based filtering for RL data curation is ubiquitous and widely accepted by the community [3, 4]. Our results already show that MATH-B’s problems are not unfair or biased. Good RL recipes (Skywork), more RL compute (Nemotron), and better distributional coverage (Qwen3 distillation) all give meaningful improvements on MATH-B. This shows that the benchmark surfaces problems that general methods can address. Further, if an exploration-oriented method succeeds in expanding several different base models, then it is likely a genuinely strong method with little risk of model-specific bias.
>
>
> **[W2 & Q2 – On larger models (>8B) and discriminative power]:** Our benchmark is designed to evaluate whether RL methods actually expand the reasoning boundary of a given model (or a set of models). The first step in such an evaluation is always to identify the subset that the base model fails on before post-training. With this framing, MATH-B is naturally targeted at academic-scale research, where models ≤8B are the ones for which both base and RL-finetuned variants are publicly available and computationally feasible to experiment with. That is, ≤8B models are precisely those that “mainstream” research on RL post-training focuses on, since experimenting with larger model sizes is only feasible for a handful of industry labs and not the broader research community.
>
>
> **[Q1 – On “genuine” exploration vs alleviating model-specific failures]:** We would also like to clarify what we mean by “exploration.” In our setting, exploration is a statistical property: does a post-training method broaden the model’s empirical distribution so that previously unsolved problems become solvable? This may involve alleviating model-specific weaknesses, but empirically that is exactly what exploration at the model level amounts to. It is about expanding the set of hard problems the model can solve. Our results in Sec. 4 already show that prolonged RL compute and improved RL recipes push models beyond their initial failure boundary. For example, with the Nemotron results (first two rows of Table 2), the core empirical question is whether an exploration-oriented RL method can push performance beyond ~8% with the same data and compute. These are exactly the kinds of methodological questions MATH-B enables us to test. We also want to acknowledge the distinction you are hinting at between statistical exploration and what might be called “genuine” reasoning progress, i.e., pushing models towards solving problems that are hard for humans as well. LLMs today are fundamentally distributional objects, and their failures often arise from subtle shifts (for example, high-digit multiplication or small variants of known problems) [5, 6, 7]. From this lens, broadening a model’s empirical competence and achieving “genuine” exploration are tightly connected. Our hope is that the academic-scale analysis enabled by MATH-B, especially in the context of exploration-driven RL, can provide insights and methods that later transfer to truly difficult problems in the style of AlphaProof or AlphaEvolve.
>
> We hope these clarifications address your concerns and help raise your confidence and rating.
>
>
> References:
>
> [1] Hendrycks et al., "Natural Adversarial Examples," 2019.
>
> [2] Goodfellow et al., “Explaining and Harnessing Adversarial Examples,” 2014
>
> [3] Albalak et al., “Big-Math: A Large-Scale, High-Quality Math Dataset for Reinforcement Learning in Language Models,” 2025
>
> [4] He et al., “DeepMath-103K: A Large-Scale, Challenging, Decontaminated, and Verifiable Mathematical Dataset for Advancing Reasoning,” 2025
>
> [5] Huang et al., “MATH-Perturb: Benchmarking LLMs' Math Reasoning Abilities against Hard Perturbations,” 2025
>
> [6] Dziri et al., “Faith and Fate: Limits of Transformers on Compositionality,” 2023
>
> [7] Mirzadeh et al., “GSM-Symbolic: Understanding the Limitations of Mathematical Reasoning in Large Language Models,” 2025

---

> > ### Comment · Reviewer_uHA5 · 2025-11-26
> >
> > Thank you for the response. I still have some concerns regarding the robustness and novelty of the paper, so I will maintain my score.

---

> > > ### Author Response · Authors · 2025-11-26
> > >
> > > Thank you for the follow-up. We had hoped our earlier response fully addressed your concerns. Could you please clarify which specific points you feel remain unresolved?

---

### Official Review · Reviewer_jrWD · 2025-11-03

**Soundness:** 3
**Presentation:** 3
**Contribution:** 3
**Rating:** 6
**Confidence:** 4

**Summary:**

The paper introduces MATH-Beyond (MATH-B), a set of small mathematical benchmarks designed to exclusively measure the Expansion Rate of post-trained LLMs, which is the percentage of problems that post-trained models can solve but base models do not even at high sampling budget (1024). The central motivation is to address the debate that current RL methods primarily sharpen existing base model skills rather than discover entirely new reasoning capabilities (Expansion).

**Strengths:**

The paper is a meaningful contribution to the community debate regarding whether RL-trained models truly discover new reasoning skills. These benchmark can provide good signal on this, as well as how to improve current methods.

The paper clearly defines the metric and is very detailed on their data filtering pipeline which makes readers more confident about the correctness of their benchmarks.

**Weaknesses:**

The resulting datasets are small enough that the authors could possibly annotate themselves, and perhaps provide further qualitative insights on these questions. In particular, why are these problems hard for the models but their GPT5-rated difficulties are lower (which the author calls  human-perceived difficulty).

I believe there are questions in the pre-filtered dataset that are rated as more difficulty (see the Omni-MATH paper, which dataset is also used here), yet somehow they end up solvable by some models, which confuses me. Could it be evidence of data leakage?

The number of benchmarked models is quite small, only 4 pairs, and all models are smaller than 8B. How do the benchmarks hold up for larger models? This would enhance the benchmark's longevity.

The authors should discuss the RL post-training data used for the post-trained models in section 4. Is there train-test leak? For example, in the pair r1-1.5b + DeepScaleR, does the training data contain some same problems as in your proposed test datasets? Similarly, the authors should discuss the SFT training data.

I'm willing to raise my score once the weaknesses are addressed.

**Questions:**

Please see my weaknesses section.

Could the author comment on the expansion rate of the 2 deepseek-r1-distill models, since they are finetuned from a teacher model as well? This is related to your paragraph lines 406 - 411. Does it make sense to include them in the base model list?

I don't understand the division into base and supplementary models and what it is supposed to show.

---

> ### Author Response · Authors · 2025-11-17
>
> We thank you for the helpful and thoughtful feedback. We address your concerns as follows:
>
> **[W1 & W2 — On human-rated vs model-specific difficulty]:** We agree that the relationship between human-perceived difficulty and model difficulty is often unintuitive. LLMs exhibit well-known distributional failure models: e.g., brittleness to high-digit arithmetic, small perturbations of familiar problems, or symbolic variants that humans find trivial [1, 2, 3]. These failures are fundamentally tied to the model’s empirical distribution rather than to human notions of conceptual hardness. In our work, we curate problems that existing base models are unable to solve (irrespective of their human-perceived difficulty), to construct a benchmark for academic-scale RL and exploration research. We examined the hardest MATH-B problems qualitatively (Tab. 5 & 6), but we did not observe clear semantic patterns explaining why these problems are difficult for models. This is consistent with prior findings showing that model hardness does not reliably map onto human difficulty ratings [1, 2, 3]. Therefore, the discrepancy you observe compared to the larger Omni-MATH dataset is expected rather than concerning.
>
> **[W3 — Longevity and evaluation on larger models]:** MATH-B is inherently model-specific: for any base model under consideration, one must first select the corresponding subset of problems that the model fails at (pass@1024 ≈ 0), and then evaluate whether post-training methods expand that boundary. This is the appropriate framing for exploration-oriented RL research. Because our target audience is academic RL research, we focus on ≤8B models, the only ones for which base and RL-tuned variants are widely available and economical to evaluate. That said, nothing in the construction prevents people from the community with available resources from applying MATH-B to larger models and finding model-specific subsets for RL research. Given that our Sec. 4 results already show that RL recipes and prolonged RL compute yield gains on these hard problems, we expect MATH-B to remain useful as a probe of capability expansion. Regarding longevity, the benchmark is not intended as a static challenge set for future models. Its purpose is to evaluate how effectively different RL methods expand the reasoning capabilities of a given base model. From this perspective, the benchmark does not become obsolete simply because new or stronger base models appear, since those improvements often stem from unrelated factors such as better distillation or larger-scale pretraining. The benchmark becomes obsolete only when substantially more powerful RL or exploration methods emerge that genuinely push base models beyond their reasoning boundary. Until that point, MATH-B serves as a clean and controlled setting for academic-scale exploration/RL research, with the potential for ideas developed here to transfer to harder frontier tasks such as those studied in AlphaProof or AlphaEvolve.
>
>
> **[W4 — On post-training data and potential leakage]:** Since MATH-B is drawn from DAPO and DeepScaleR, some overlap with the training sets of ProRL/Nemotron or DeepScaleR models is possible. However, the key point is that MATH-B problems are selected because the corresponding base models fail them at pass@1024. This makes it unlikely that these samples were meaningfully used for policy-loss updates during post-training, although they may contribute minimally through KL/entropy terms. For DeepScaleR in particular, the training regime (about 90 epochs with a group size of 8 [4]) corresponds to roughly pass@720 exposure. Given the limited expansion capability observed in our results and in prior work [5,6], it is unlikely that these questions were effectively learned during training. This is also supported by Table 2, where DeepScaleR reaches only 5.22% pass@1024. The fact that this is pass@1024 rather than pass@1 indicates that even with extensive sampling the success rate remains extremely low. If any leakage exists, its effect is therefore very small. The same reasoning applies to the Nemotron models.
>
>
> **[Q1 & Q2 — On base & supp. models]:** Our terminology is purely functional. Base models are those that are typically used by practitioners for RL-based reasoning research (e.g., Qwen2.5, DeepSeek-R1-Distill). Although r1-distill is itself a distilled model, it is widely used as a base for RL; thus we include it. Supplementary models are the corresponding post-trained variants used to illustrate whether expansion occurs. This division allows us to (1) construct the benchmark using base models, and (2) illustrate expansion using matched post-trained models. Nothing hinges on the naming; it simply reflects common usage patterns in the community. That said, a model may appear on both sides depending on context: the categorization is a convenience, not a conceptual claim.
>
>
> We hope this addresses your concerns fully and encourages you to raise your rating and confidence.

---

> > ### Author Response · Authors · 2025-11-17
> >
> > References:
> >
> > [1] Huang et al., “MATH-Perturb: Benchmarking LLMs' Math Reasoning Abilities against Hard Perturbations,” 2025
> >
> > [2] Dziri et al., “Faith and Fate: Limits of Transformers on Compositionality,” 2023
> >
> > [3] Mirzadeh et al., “GSM-Symbolic: Understanding the Limitations of Mathematical Reasoning in Large Language Models,” 2025
> >
> > [4] https://github.com/rllm-org/rllm/tree/main/scripts/train/deepscaler_1.5b
> >
> > [5] Yue et al., “Does Reinforcement Learning Really Incentivize Reasoning Capacity in LLMs Beyond the Base Model?,” 2025
> >
> > [6] Wu et al., “The Invisible Leash: Why RLVR May or May Not Escape Its Origin,” 2025

---

### Meta-Review · Area_Chair_z7Aw · 2026-01-04

**Summary:**

- **Summary**:The paper introduces MATH-B, a benchmark constructed from hard instances in DAPO-Math-17K and DeepScaleR (unsolved at pass@1024) to evaluate if RL models truly outperform their base versions. This work significantly contributes to measuring the expansion of model capability boundaries.
- **Pros**: The proposed method for evaluating the effectiveness of RL/exploration in expanding reasoning boundaries is novel. The authors also presented the correlation between the benchmark scores and the RL algorithms (like Nemotron), providing further evidence of its validity.
- **Cons**: The evaluation is limited to models $\leq$ 8B. This lack of validation on larger models limits the persuasiveness of the claims. Also, the absence of new questions and fine-grained annotations—such as difficulty levels—limits the comprehensiveness of the dataset, rendering the benchmark construction less sophisticated than expected.

Despite the limitation in model scale, I recommend acceptance due to the paper's novel methodology and interesting findings.

**Reviewer Concerns:**

- **Addressed Concerns**: The rebuttal likely clarified several specific technical inquiries. I believe the authors have adequately responded to Reviewer `jrWD` and Reviewer `aWPb` regarding the discrepancy between model performance and human-perceived difficulty (GPT-5 ratings), as well as the specific concerns regarding data leakage and train-test overlap raised by Reviewer `jrWD`. Additionally, the technical clarifications regarding the metadata of comparison datasets (e.g., AIME) raised by Reviewer `VmSx` is also addressed by figure updated.
- **Outstanding Concerns**:
  1. Limited Model Scope: Multiple reviewers (`jrWD`, `uHA5`) emphasized that evaluating only models $\leq$ 8B parameters limits the benchmark's longevity and persuasiveness. The lack of validation on larger $\geq$ 70B models leaves the benchmark's effectiveness at scale unverified.
  2. Simple construction pipeline: Reviewer `aWPb` raised a fundamental concern that the proposed benchmark is essentially a subset of existing datasets (DAPO, DeepScaleR) without introducing new problem formulations or evaluation methodologies. The paper confirms known limitations of RL rather than offering novel insights.

**Reviewer Scores:**

- Reviewer `jrWD`: This reviewer likely maintained the score, as the core limitation regarding the exclusive evaluation of small-scale models (<8B) persists despite technical clarifications.

- Reviewer `uHA5`: This reviewer would likely maintain the score due to outstanding concerns about circular bias in data filtering and the lack of validation on larger models.

- Reviewer `VmSx`: This reviewer likely maintained the score given the continued lack of empirical evidence demonstrating the dataset's practical utility for improving RL training.

- Reviewer `aWPb`: This reviewer likely maintained the score, as the fundamental critique regarding the limited construction methods and new questions are not proposed in this benchmark.

---

### Decision · Program_Chairs · 2026-01-26

Accept (Poster)